# Precision Therapeutic and Preventive Molecular Strategies for Endometriosis-Associated Infertility

**DOI:** 10.3390/ijms26167706

**Published:** 2025-08-09

**Authors:** Inès Limam, Mohamed Abdelkarim, Khadija Kacem-Berjeb, Mohamed Khrouf, Anis Feki, Marouen Braham, Nozha Chakroun

**Affiliations:** 1LR99ES11, Faculty of Medicine of Tunis, Tunis El Manar University, Tunis 1006, Tunisia; ines.limam@fmt.utm.tn; 2Population Health, Environmental Aggressors, Alternative Therapies, Faculty of Medicine of Tunis, Tunis El Manar University, Tunis 1006, Tunisia; mohamed.abdelkarim@fmt.utm.tn; 3Department of Reproductive Medecine, Aziza Othmana University-Hospital, Health Ministry, Tunis 1008, Tunisia; khadija.berjeb@gmail.com (K.K.-B.); braham.marouen@gmail.com (M.B.); 4LR16SP01 Procréation-Infertilité, Aziza Othmana University-Hospital, Tunis El Manar University, Tunis 1008, Tunisia; moh.khrouf@gmail.com; 5Centre FERTILLIA de Médecine de la Reproduction-Clinique la Rose, Tunis 1053, Tunisia; 6Department of Gynecology and Obstetrics, University Hospital of Fribourg, 1708 Fribourg, Switzerland; anis.feki@h-fr.ch

**Keywords:** endometriosis-associated infertility, epigenetics, biomarkers, prevention, therapeutics

## Abstract

Endometriosis, a chronic estrogen-dependent disorder defined by ectopic endometrial-like tissue growth, causes pelvic pain and infertility in reproductive-age women. Despite its prevalence, the underlying mechanisms driving lesion persistence and reproductive impairment remain unclear. This review synthesizes recent pathophysiological advances, highlighting how hormonal dysregulation, immune dysfunction, epigenetic alterations, and oxidative stress collectively foster lesion persistence and treatment resistance. Critically, these molecular disturbances disrupt critical reproductive functions—including oocyte quality, endometrial receptivity, and embryo implantation. We further explore emerging non-hormonal therapeutic strategies, including MAPK and PI3K/AKT inhibitors as well as epigenetic agents targeting HOXA10 methylation and microRNA modulation, which offer fertility-sparing alternatives to conventional hormonal suppression. To enhance clinical translation, we propose a multi-level prevention framework—encompassing at the primary level, risk reduction; at the secondary level, biomarker-guided intervention; and at the tertiary level, fertility preservation—to anticipate disease progression and personalize reproductive care. By delineating shared pathways between endometriosis and infertility, this work advances precision medicine approaches for affected patients.

## 1. Introduction

Endometriosis—a chronic inflammatory disorder driven by estrogen-dependent growth of ectopic endometrial tissue—remains a formidable clinical challenge, profoundly impacting the physical, psychological, and reproductive health of 190 million women worldwide [1,2]. Despite affecting approximatively 10% of reproductive-age women and up to 50% of infertility cases, its elusive pathogenesis results in diagnostic delays averaging 7–10 years and limited therapeutic efficacy beyond symptomatic control [3]. Classical theories about its pathogenesis—such as retrograde menstruation, metaplasia, and benign metastasis—have laid important foundations, but recent molecular insights have revealed a far more complex systemic disorder [4]. Crucially, molecular dysregulation lies at the heart of this disease. Aberrant estrogen signaling, progesterone resistance, and immune dysfunction collaborate to create a self-sustaining microenvironment that promotes lesion survival, angiogenesis, and therapeutic evasion [5,6]. These same mechanisms directly impair key reproductive functions and contribute to infertility, not merely through anatomical distortion, but by disrupting oocyte competence, endometrial receptivity, and embryo–maternal crosstalk. Even the mildest disease can inflict molecular scars that compromise pregnancy chances.

This review deciphers how endometriosis hijacks reproductive biology at the molecular level. We first unravel the pathogenic triad (hormonal, immune, and epigenetic disturbances) fueling lesion persistence. We then expose their direct effects on fertility—from folliculogenesis to implantation. By connecting these dots, we highlight actionable therapeutic windows and advocate for precision care tailored to women’s reproductive aspirations.

## 2. Molecular Mechanisms of Endometriosis

The pathogenesis of endometriosis is driven by a complex interplay of hormonal, immune, oxidative, and epigenetic dysregulation. Estrogen dominance and progesterone resistance sustain lesion growth, while chronic inflammation—mediated by interleukin (IL)-1β and tumor necrosis factor (TNF)-α—amplifies tissue injury. Deficient natural killer (NK) cell surveillance, oxidative DNA damage, angiogenic remodeling, and epithelial-to-mesenchymal transition (EMT) enhance immune evasion and invasiveness. Together, these dysregulated processes form a self-reinforcing loop, perpetuating lesions and impairing fertility. The subsections below detail the key molecular pathways in this pathogenic cascade.

### 2.1. Hormonal Dysregulation: Estrogen Dominance and Progesterone Resistance

Endometriotic implants acquire biochemical autonomy, behaving like ectopic endocrine units that escape normal ovarian regulation. This autonomy arises from a combination of enzyme induction, receptor alteration, and epigenetic dysregulation, which collectively drive local estrogen excess and progesterone resistance [7].

Estrogen synthesis is amplified through two main routes. First, prostaglandin-E_2_ (PGE_2_) stimulates steroidogenic factor-1 (SF-1), enhancing Cytochrome P450 19A1 (CYP19A1; aromatase) expression and creating a feedback loop with estradiol and cyclo-oxygenase-2 [8,9,10]. Second, a sulfatase pathway complements de novo synthesis: steroid sulfatase releases estrone from circulating estrone-sulfate imported via the transporter solute carrier family 10 member 6 (SLC10A6), ensuring precursor availability even in low androgen conditions [11,12,13]. Simultaneously, reduced 17β-hydroxysteroid dehydrogenase type 2 (17β-HSD2) activity limits estradiol inactivation, prolonging estrogenic signaling [14,15].

Estrogen-mediated responses are also altered. A shift toward estrogen receptor β (ERβ) over ERα in stromal cells promotes inflammation and inhibits apoptosis, supporting lesion persistence [16,17,18,19,20]. ERβ also activates the NACHT, LRR and PYD domains-containing protein 3 (NLRP3) inflammasome via caspase-1, increasing IL-1β and linking estrogen signaling to sterile inflammation [21,22,23]. In parallel, cytoplasmic ERβ disrupts TNF-α-induced apoptosis by interacting with Apoptosis signal-regulating kinase 1 (ASK1), serine/threonine kinase receptor-associated Protein (STRAP), and 14-3-3 proteins [24,25,26]. Additionally, G-protein-coupled estrogen receptor (GPER, also known as GPR30) accelerates estrogen-driven signaling by mobilizing cAMP and transactivating SF-1, further boosting aromatase transcription [27,28,29,30]. This may explain the responsiveness to xenoestrogens such as bisphenol A (BPA) and Selective Estrogen Receptor Modulators (SERMs) like tamoxifen.

Progesterone resistance is largely epigenetic. Promoter hypermethylation and histone deacetylase (HDAC)-mediated repression of the progesterone receptor gene reduce progesterone receptor isoform B (PR-B) expression, a hallmark of resistance [31,32,33,34]. Estrogen contributes by upregulating DNA methyltransferase 1 (DNMT1), silencing Runt-related transcription factor (RUNX3) via promoter hypermethylation [35]. BPA reinforces this through a WD repeat domain 5 (WDR5)/ten–eleven translocation methylcytosine dioxygenase 2 (TET2) axis, which increases ERβ transcription [36].

Metabolic–epigenetic crosstalk has recently emerged as a further amplifier. Cellular prion protein (PrPC), an estrogen-induced gene, represses peroxisome proliferator-activated receptor-α (PPAR-α), driving cholesterol accumulation and further boosting aromatase activity [37]. Estradiol then increases PrPC levels, closing a self-perpetuating metabolic loop [38,39,40].

Collectively, enhanced enzymatic activity, receptor remodeling, and epigenetic changes generate a persistent estrogen-dominant, progesterone-resistant environment that initiates and sustains downstream cellular dysregulation.

### 2.2. Epigenetic Reprogramming and Non-Coding RNAs

Endometriosis is characterized by persistent transcriptional rewiring that is largely driven by epigenetic and post-transcriptional mechanisms. These alterations precede lesion visibility and shape the hormonal, inflammatory, and immune landscape of affected tissues.

While numerous epigenetic regulators have been described, Table 1 summarizes a selected subset—primarily those with documented effects on genes involved in endometrial receptivity, hormone sensitivity, or immune modulation. These elements are particularly relevant for infertility and will be further discussed as potential therapeutic targets in later sections.

This regulatory framework lays the molecular foundation upon which chronic signaling cascades operate, notably those involved in intracellular survival, proliferation, and invasion, as explored in the following subsection.

### 2.3. Intracellular Signaling Pathways

Endometriotic cells operate within a reprogrammed intracellular environment, shaped by constitutively active signaling pathways that reinforce their survival, invasiveness, and resistance. Two core axes—the PI3K/AKT/mTOR and Wnt/β-catenin pathways—form the backbone of this deregulated signaling network [58,59,60,61,62].

Hyperactivated PI3K/AKT/mTOR signaling enhances glucose uptake, stimulates aerobic glycolysis, promotes angiogenesis, and weakens immune detection [63,64,65]. It also impairs progesterone signaling, deepening the hormonal resistance initiated by epigenetic silencing [66].

In parallel, dysregulated Wnt/β-catenin signaling triggers nuclear translocation of β-catenin, which partners with TCF/LEF transcription factors to activate genes involved in motility, extracellular matrix remodeling, and EMT (epithelial–mesenchymal transition), including MMP-2 and MMP-9 [61,67].

These pathways intersect with the TGF-β axis, a central orchestrator of EMT and fibrosis. SMAD complexes, activated downstream of TGF-β, cooperate with β-catenin and phosphorylated AKT to upregulate SNAIL, SLUG, and ZEB1/2; repress E-cadherin; and induce vimentin and N-cadherin—hallmarks of increased cell motility and reduced adhesion [68].

Altogether, PI3K/AKT, Wnt/β-catenin, and TGF-β signaling form an integrated circuit that processes hormonal, inflammatory, and metabolic cues. This molecular machinery confers plasticity and invasiveness, transforming endometrial cells into a resilient, immune-evasive phenotype that thrives in the hostile peritoneal microenvironment.

### 2.4. Inflammation, Immune Cell Recruitment, and Neuro-Immune Crosstalk

Endometriotic lesions develop within a chronic inflammatory microenvironment shaped by oxidative stress, cytokine overload, and immune cell infiltration. In addition to fueling local proliferation, these cues recruit both innate and adaptive immune cells and initiate neurogenic pathways that contribute to pain and tissue remodeling. The peritoneal fluid of women with endometriosis is enriched in activated macrophages, dendritic cells, mast cells, neutrophils, and CD4+/CD8+ lymphocytes, all of which infiltrate ectopic sites and amplify cytokine production [69,70,71]. Single-cell RNA sequencing (scRNA-seq) has recently identified a macrophage subset within lesions that resembles tumor-associated macrophages. These cells suppress cholesterol efflux via the ATP-binding cassette transporter A1 (ABCA1) and ATP-binding cassette transporter G1 (ABCG1) transporters, accumulate intracellular lipids, and enhance IL-1β, IL-6, and PGE_2_ synthesis [72]. The upstream regulators liver X receptor (LXR) and peroxisome proliferator-activated receptor (PPAR), which modulate lipid metabolism, emerge as potential therapeutic targets.

Danger signals such as hypoxia and oxidative cell debris further promote IL-1β, IL-6, IL-8, TNF-α, and IL-17A release. IL-17A, in particular, promotes vascular endothelial growth factor (VEGF) transcription and enhances MMP production, and when neutralized, reduces vascularization and lesion size in vivo [73,74,75,76]. In parallel, IL-1β and TNF-α reinforce inflammation by upregulating cyclo-oxygenase-2 (COX2) and stimulating PGE2 production, thus sustaining local estrogen biosynthesis [30,77,78,79].

Complement signaling is also implicated. Activation of the c5a receptor (C5aR1) on macrophages and Schwann cells boosts inflammasome responses and sensitizes pain pathways [80]. Deep lesions also contain tryptase-positive mast cells, which activate protease-activated receptor-2 (PAR-2), promote fibrosis, and sustain neurogenic inflammation [81].

These inflammatory and immune signals converge on the TGF-β axis. TGF-β-induced SMAD complexes, in cooperation with AKT and β-catenin, upregulate SNAIL, SLUG, and ZEB1/2 while repressing E-cadherin. This promotes fibroblast-to-myofibroblast conversion, collagen deposition, and matrix stiffening [82,83]. TGF-β also expands FOXP3+ regulatory T cells and inhibits cytotoxic immune responses, creating a tolerogenic niche [84,85,86].

Neuro-immune crosstalk further supports lesion persistence. Inflammatory cytokines stimulate the release of nerve growth factor (NGF) and brain-derived neurotrophic factor, leading to sensory nerve sprouting and heightened nociceptor sensitivity [87]. The resulting peritoneal niche is thus proinflammatory, neurotrophic, and angiogenic—favoring lesion survival, invasion, and pain.

### 2.5. Oxidative Stress and Peritoneal Toxicity

Retrograde menstruation and lesion hemorrhage expose the peritoneal cavity to hemolyzed erythrocytes, triggering iron release through macrophage-mediated clearance. This overwhelms iron-binding proteins and initiates Fenton reactions that produce hydroxyl radicals. Elevated levels of ferritin, transferrin, and free iron in the peritoneal fluid coincide with increased lipid peroxides and protein carbonyls, markers of oxidative injury [88,89,90,91,92].

This oxidative stress compromises the mitochondrial membrane potential and disrupts electron transport while promoting cytochrome c release, impairing DNA repair mechanisms [93,94,95]. Damaged macromolecules serve as danger-associated molecular patterns (DAMPs) that further activate macrophages and reinforce inflammation [96].

Beyond damage, reactive oxygen species (ROS) act as epigenetic modulators, altering DNA methylation at antioxidative gene loci via Ten–Eleven Translocation (TET) dioxygenase enzymes and supporting pro-survival transcriptional programs [97]. Systemically, ROS damage granulosa cells, impair the mitochondrial distribution [98], deplete glutathione, and destabilize the meiotic spindle—key contributors to poor oocyte quality [99].

In the peritoneal niche, oxidative damage also remodels the extracellular matrix by modifying structural proteins and activating redox-sensitive proteases, facilitating cellular invasion—a process expanded in the following section.

### 2.6. Extracellular Matrix Remodeling, EMT, and Fibrogenesis

Extracellular matrix (ECM) remodeling in endometriosis represents a dynamic interplay between degradation and fibrotic reconstruction. Rather than detailing each molecular cascade, we have summarized the key pathways and mediators involved in ECM breakdown, EMT, and fibrogenesis in the table below (Table 2). These events collectively remodel the lesion microenvironment into a fibrotic and immune-evasive niche that fosters lesion survival and impairs surgical outcomes. Notably, several of the implicated pathways—such as TGF-β, lysyl oxidase (LOX)/LOX-like, and urokinase-type plasminogen activator receptor (uPAR)—overlap with angiogenic and immune signaling, reinforcing their central role in disease progression.

### 2.7. Aberrant Vascular Remodeling and Non-Canonical Angiogenic Pathways

Endometriotic angiogenesis diverges markedly from physiological endometrial vascularization. Hypoxia and oxidative stress stabilize hypoxia-inducible factor-1 α (HIF-1α), which cooperates with inflammatory cytokines to induce VEGF-A expression. However, the resulting microvessels are structurally abnormal—disorganized, leaky, and poorly covered by pericytes [113,114]. Additional non-canonical pathways—including slit guidance ligand 2 (Slit2)/Roundabout guidance receptor 1 (ROBO1), prokineticin-1 (PROK1)/prokineticin-1 receptor (PROKR1), and Notch/delta-like ligand 4 (Dll4) suppression— further disrupt vessel patterning and branching, as detailed in Figure 1 [115,116,117,118]. This abnormal vascular network perpetuates hypoxia, prolongs HIF-1α activation, and creates a permissive niche for stromal invasion, immune infiltration, and nerve fiber ingrowth. These features reinforce chronic inflammation and pain, forming an integral part of the self-sustaining inflammatory loop that promotes lesion persistence and compromises reproductive function.

## 3. Molecular Bridges Between Endometriosis and Infertility

Infertility in endometriosis reflects a progressive, system-wide dysfunction that impairs every stage of the reproductive process—from early follicular development to embryo implantation. Cytokines, iron-induced oxidative stress, hormonal imbalance, and epigenetic changes extend beyond visible lesions, altering the molecular landscape of tissues that appear normal upon imaging or laparoscopy. Recent advances in single-cell mapping, metabolomics, and spatial transcriptomics reveal a shared signature of inflammation, oxidative damage, and hormonal stress across the ovary, fallopian tube, and uterus [119]. Figure 2 illustrates six interconnected mechanisms that collectively compromise reproductive function.

### 3.1. Ovarian Folliculogenesis and Reserve Erosion

The ovarian environment in endometriosis is exposed to chronic inflammation and oxidative stress, which, together, disrupt folliculogenesis and accelerate reserve depletion. Cytokines such as TNF-α and IL-1β impair granulosa cell function, downregulating key genes, including *FSH receptor* (*FSHR*) and aromatase (*CYP19A1*), which are essential for follicle maturation and estrogen synthesis [120,121,122,123,124]. Endometriomas exert both mechanical pressure on and biochemical damage to the surrounding ovarian cortex. Their cystic fluid—rich in free iron, proteases, and ROS—induces stromal fibrosis, follicular apoptosis, and premature activation of primordial follicles, gradually exhausting the reserve. Clinically, this manifests as lower serum anti-Müllerian hormone (AMH) and reduced antral follicle counts, two markers of diminished ovarian function [125,126]. Although surgical excision of endometriomas may be necessary, especially for large or symptomatic lesions, it risks removing healthy tissue, further reducing the ovarian reserve. This underscores the need for careful preoperative evaluation in reproductive-age women and supports fertility-preserving options, such as cyst-sparing surgery, ovarian tissue cryopreservation, or medical suppression before and after intervention [127].

### 3.2. Oxidative Damage to Oocytes and Early Embryos

Oxidative stress in endometriosis directly compromises oocyte integrity. Exposure to pathological levels of ROS (≈150 µM H_2_O_2_) disrupts mitochondrial distribution, disorganizes the actin–microtubule network, and impairs kinetochore alignment, reducing fertilization [128,129,130,131,132,133,134]. Consequently, early embryos from affected women show increased fragmentation and developmental arrest, reflecting residual oxidative and spindle damage [135].

### 3.3. Tubal Ciliary Dysfunction and Peritoneal Toxicity

In endometriosis, inflammation spreads beyond lesions, affecting the fallopian tubes and peritoneal cavity and disrupting fertilization and early embryo transport. Elevated levels of IL-6 and TNF-α impair both ciliary motion and smooth muscle contraction, two essential mechanisms for gamete capture and embryo transit [136,137,138]. Chronic exposure to cytokine and prostaglandin damages the ciliated epithelium and reduces tubal motility, further compromising transport [138,139]. Fibrosis and adhesions form physical obstacles that block sperm–oocyte interactions and hinder embryo passage. The resultant peritoneal fluid is rich in cytokines, proteases, and ROS, which creates a toxic microenvironment that harms fertilization and early embryonic development [140,141,142].

### 3.4. Endometrial Receptivity Breakdown

Because the embryo reaches the uterus just days after fertilization, the brief “window of implantation” forms a critical link between fertilization, blastocyst development, and uterine attachment. Disruption at this stage can quickly compromise fertility. Successful implantation requires a properly primed endometrium [143], yet in women with endometriosis, key receptivity markers undergo epigenetic silencing [144,145]. Among them, the transcription factor HOXA10 plays a central role in stromal differentiation and integrin expression. Its silencing reduces integrin αvβ3, leukemia inhibitory factor (LIF), and glycodelin—all essential for embryo adhesion [44,146,147]. Transcriptomic analyses show that the overexpression of miR-135a/b reinforces HOXA10 silencing, intensifying the epigenetic brake on implantation competence [45]. Concurrently, diminished PR-B expression limits stromal decidualization and contributes to progesterone resistance [148,149]. These changes are reinforced by the aberrant activation of PI3K/AKT and mitogen-activated protein kinase (MAPK) pathways, which further impair endometrial receptivity [150,151].

### 3.5. Maternal–Fetal Immune Imbalance and Implantation Failure

Once the blastocyst attaches to a receptive endometrium, the maternal immune system must swiftly shift from surveillance to tolerance at the maternal–fetal interface. In endometriosis, failure of this transition represents a second major bottleneck contributing to implantation failure [152,153]. Dysfunctional uterine natural killer (uNK) cells exhibit altered cytokine secretion and reduced angiogenic capacity, compromising spiral artery remodeling and early placental development [154]. Concurrently, deficient regulatory T cells (Tregs) disrupt maternal immune tolerance, increasing the risk of embryo rejection [155,156]. A skew toward pro-inflammatory Th17 cells—driven by an upregulated IL-23/Th17 axis—exacerbates oxidative stress and destabilizes the implantation micro-environment [157]. Clinical studies in women with repeated implantation failure confirm an elevated Th17/Treg ratio and heightened endometrial inflammation [158], highlighting the immunologic barrier to successful nidation.

### 3.6. Genetic Susceptibility and Metabolic Rewiring

Even after fertilization and a brief immune truce, genetic predisposition and metabolic disruption can still compromise embryonic development and placentation [159]. Genome-wide association studies (GWAS) have identified susceptibility loci such as *wnt family member 4 (WNT4)*, *Vezatin adherens junctions transmembrane protein (VEZT)*, and *cyclin-dependent kinase inhibitor 2B antisense RNA 1 (CDKN2B-AS1)*, which shape uterine development, endometrial receptivity, and cell proliferation [160,161,162]. Endometriotic lesions rewire metabolism, especially tryptophan and arginine pathways, which fuel immune imbalance, oxidative stress, and chronic inflammation [163,164,165]. Targeted metabolomic analysis of follicular fluid has revealed disruptions in sphingolipid and ceramide pathways, correlating with poorer in vitro fertilization (IVF) outcomes [166]. In the eutopic endometrium, reduced LIF–signal transducer and activator of transcription 3 (STAT3) activity and weakened progesterone response impair implantation, especially in women with repeated failure [167]. These findings connect immune, hormonal, and metabolic stress into a common pathway that undermines fertility.

## 4. Emerging Therapeutic Targets in Endometriosis-Associated Infertility

Recent advances in the molecular basis of endometriosis have fueled the development of targeted therapies aimed at preserving or restoring fertility rather than just managing symptoms. This shift marks a move from traditional symptom-oriented approaches to mechanism-based interventions that directly address the root causes of lesion persistence and reproductive dysfunction [168]. Several emerging non-hormonal approaches now focus on key molecular pathways—such as inflammation, oxidative stress, angiogenesis, and immune imbalance—that drive disease progression (Figure 3).

### 4.1. Targeting Estrogen Receptor Beta (ERβ) Signaling

Endometriotic lesions consistently exhibit a dysregulated estrogen receptor profile, marked by a heightened ERβ-to-ERα ratio. This receptor imbalance fosters a microenvironment of sustained inflammation, apoptotic resistance, progesterone insensitivity, and angiogenesis. Through transcriptional reprogramming, ERβ activation promotes lesion survival and impairs receptivity. Selective ERβ antagonists have shown promising results in preclinical models, reducing lesion size and partially restoring progesterone responsiveness [169,170]. Unlike systemic estrogen suppression, these agents modulate local estrogen signaling without compromising ovulation, making them particularly attractive for fertility-preserving interventions.

In parallel, intracrine steroidogenesis and receptor-subtype modulation are being explored to rebalance endometrial hormonal signaling. Selective estrogen receptor modulators (SERMs), selective androgen receptor modulators (SARMs), and selective androgen receptor degraders (SARDs) target pathways that interact with enzymes such as aromatase, steroid sulfatase, and aldo-keto reductase family 1 member C3 (AKR1C3), which regulate local estrogen and androgen biosynthesis. By modulating hormonal, immune, and vascular crosstalk, these therapies aim to reestablish a receptive endometrial microenvironment and improve fertility outcomes [171].

Taken together, ERβ-targeted therapies offer a mechanistically precise, fertility-friendly alternative to conventional hormone suppression.

### 4.2. Inhibition of Inflammatory Kinase Pathways

Key intracellular kinase cascades—NF-κB, MAPK, and PI3K/AKT/mTOR—are persistently activated in endometriotic lesions, sustaining inflammation, proliferation, and progesterone resistance [172,173,174,175]. NF-κB enhances cytokine production and immune evasion, while the MAPK axis (extracellular signal-regulated kinases 1 and 2 (ERK1/2), c-Jun N-terminal kinase (JNK, p38) promotes epithelial growth and matrix degradation. The PI3K/AKT/mTOR pathway fuels stromal expansion, neovascularization, and hormonal insensitivity.

Pharmacological inhibition of these kinases shows promise in preclinical studies. Mitogen-acctivated protein kinase kinase (MEK) inhibitors (e.g., U0126) and NF-κB pathway blockers (e.g., BAY11-7082) reduce lesion volume and inflammatory infiltration, downregulating proliferating cell nuclear antigen (PCNA) and MMP9 expression [176]. mTOR inhibitors such as everolimus and rapamycin—already used in oncology—also suppress lesion growth and improve implantation outcomes, especially when combined with hormonal agents [177].

Oxidative stress appears to amplify kinase activity, partly via deubiquitinase cylindromatosis (CYLD), a redox-sensitive inhibitor of NF-κB [178], further linking inflammation to redox imbalance. Taken together, kinase inhibitors represent a promising class of non-hormonal agents capable of curbing lesion progression, restoring endocrine responsiveness, and enhancing the fertility potential.

### 4.3. Non-Hormonal Immunomodulation and Immune-Checkpoint Targets

Immune dysfunction in endometriosis involves chronic inflammation, impaired clearance, and local tolerance that permits ectopic lesion survival. Immune checkpoints, particularly programmed cell death protein 1 (PD-1) and its ligand PD-L1, have emerged as novel targets for restoring immune surveillance, reducing inflammation, and enhancing endometrial receptivity [179,180]. Similarly, the CD47–signal-regulatory protein alpha (SIRPα) axis allows stromal cells to evade macrophage-mediated phagocytosis. Blocking CD47 enhances macrophage-mediated clearance and induces ectopic cell apoptosis [181].

Non-hormonal immunomodulators also show therapeutic promise. TNF-α inhibitors such as etanercept and infliximab have demonstrated efficacy in reducing lesion volume and pelvic pain in animal models, while recombinant TNFRSF1A (r-hTBP1) improves outcomes without affecting estrogen signaling, supporting fertility preservation [182,183,184]. IL-8 blockade with AMY109 has shown a favorable safety profile in Phase 1 trials and does not interfere with menstrual cycling [185]. However, some cytokine-targeted therapies, such as interferon-α2b, have failed to show benefit and may increase the recurrence risk [186].

### 4.4. Epigenetic Therapies and Non-Coding RNA Modulators

Epigenetic remodeling plays a crucial role in disrupting progesterone responsiveness and endometrial receptivity in endometriosis-associated infertility. Aberrant DNA methylation and histone deacetylation frequently silence genes such as HOXA10 and PR-B, which are both essential for stromal decidualization [34,187,188,189]. By reversing these repressive marks, HDAC inhibitors and demethylating agents like 5-azacytidine have shown potential for restoring implantation-related gene expression in vitro [190].

In parallel, non-coding RNAs—especially miRNAs—modulate a broad range of pathological pathways, from inflammation to fibrosis. Several miRNAs dysregulated in endometriosis, including let-7b, miR-135a/b, and miR-29c, influence gene networks involved in hormone signaling and endometrial remodeling [191]. In preclinical models, silencing of miR-451a reduced lesion size and downregulated targets such as 14-3-3 protein zeta/delta (YWHAZ), MAPK1, β-catenin (CTNNB1), and IL-6 [192]. Other regulatory axes—like circ_0007331/miR-200c-3p/HIF-1α—have been shown to drive angiogenesis and invasion, and their inhibition curbs disease progression [193]. Fibrotic remodeling, a key feature of deep infiltrating lesions, can also be alleviated via exosomal delivery of miR-214, which suppresses connective tissue growth factor (CTGF) and collagen type I alpha 1 chain (COL1A1) [194]. Additionally, miR-205-5p was reported to limit stromal invasion by targeting angiopoietin-2 (ANGPT2) and dampening AKT/ERK signaling, with lower expression levels correlating with greater disease severity in patient samples [195]. These molecular findings support the rationale for developing RNA-based and epigenetic therapies as precision tools to correct aberrant signaling and improve reproductive outcomes without relying on systemic hormonal suppression.

### 4.5. Antioxidant-Based Strategies and Redox Modulation

Oxidative stress is increasingly being recognized as a key contributor to the reproductive complications observed in endometriosis, including oocyte damage, impaired fertilization, and embryo developmental arrest. Elevated levels of ROS disrupt mitochondrial function, impair chromosomal alignment, and activate inflammatory pathways such as MAPK and NF-κB, aggravating hormonal resistance [95].

In response, antioxidative therapies have gained attention as adjunct strategies to restore redox balance and support fertility. Early studies suggest that vitamins C and E can reduce the levels of oxidative markers and improve oocyte quality, especially in the context of assisted reproductive technologies (ART), where the oxidative burden is elevated [196,197]. Coenzyme Q10 has also shown promise by supporting mitochondrial bioenergetics and supporting embryo development under oxidative stress [196]. Melatonin, another potent antioxidant, improves mitochondrial function and reduces inflammation in preclinical endometriosis models [198,199,200].

Among glutathione precursors, N-acetylcysteine (NAC) has demonstrated both anti-inflammatory and fertility-enhancing effects, improving oocyte quality and fertilization rates in IVF settings [201]. Plant-derived polyphenols exhibit a broad spectrum of bioactive properties, including antioxidative, anti-inflammatory, antiproliferative, and anti-angiogenic effects across various disease contexts, such as cancer and reproductive disorders [202,203,204,205]. Among them, resveratrol has been particularly studied in endometriosis for its ability to enhance endometrial receptivity and reduce lesion vascularization, contributing to better implantation outcomes [204,206,207].

Despite limited clinical data, these compounds represent promising adjuncts for managing oxidative-stress-related infertility in endometriosis. However, further studies are needed to confirm their impact on reproductive outcomes [208].

### 4.6. Anti-Angiogenic and Vascular Normalization Therapies

Aberrant angiogenesis is central to endometriotic lesion growth, ensuring vascular support for ectopic tissue survival and invasion. This process is largely driven by the overexpression of VEGF and its receptor VEGFR2 in the peritoneal environment [209]. Thus, targeting this axis offers a non-hormonal strategy to halt disease progression.

In preclinical models, VEGF inhibitors such as aflibercept, a recombinant VEGF trap, significantly reduced lesion volume lesion size, VEGF expression, and microvessel density, outperforming leuprolide acetate in relation to several histological markers [210]. In parallel, the bradykinin B1 receptor antagonist R-954 decreased VEGF/VEGFR expression, cyst formation, cytokine production, and inflammatory infiltration without disturbing estrous cyclicity, indicating its fertility-friendly profile [211,212].

Dopamine agonists like cabergoline and quinagolide also block the VEGF/VEGFR2 pathway. In women with endometriosis-associated hyperprolactinemia, quinagolide reduced both lesion vascularization and overall lesion volume [213,214].

Table 3 summarizes the most promising fertility-oriented therapeutic candidates under investigation, highlighting their mechanisms of action and translational potential.

Despite their therapeutic promise, several barriers limit their clinical implementation. These include suboptimal pharmacokinetics, delivery challenges, and concerns about long-term safety and off-target effects—particularly for RNA-based agents [218,219,220,221]. Regulatory frameworks also require extensive toxicological and immunogenicity testing before approval [222,223]. Moreover, the high cost of development and underinvestment in non-hormonal endometriosis research continue to hamper fertility-focused clinical trials [224,225]. Overcoming these hurdles will require coordinated investment, regulatory innovation, and trial designs that prioritize reproductive outcomes.

## 5. From Risk Reduction to Fertility Preservation: Biomarker-Guided Strategies in Endometriosis Care

Despite major advances in the molecular understanding of endometriosis, preventive strategies—particularly those aiming to preserve fertility—remain insufficiently developed. Incorporating biomarkers into preventive care offers a paradigm shift toward earlier identification of at-risk individuals and improved reproductive outcomes [226]. Yet, no single biomarker has demonstrated adequate accuracy or disease specificity to be used as a standalone diagnostic or triage tool. Many candidates are non-specific and also elevated in other inflammatory or physiological states, leading to their limited clinical use.

Recent developments in high-throughput omics technologies—including proteomics, transcriptomics, and metabolomics—have accelerated the discovery of clinically relevant biomarkers [227]. This section highlights their emerging role across three levels of prevention: stratifying risk before symptom onset (primary), preserving ovarian function in early-stage disease (secondary), and personalizing reproductive management after treatment (tertiary) [228].

### 5.1. Primary Prevention: Risk Stratification and Early Identification

Primary prevention targets disease onset in asymptomatic individuals, particularly those with familial predisposition or early-life risk factors. Well-known risk factors include early menarche, short menstrual cycles, nulliparity, and a family history—all of which contribute to prolonged estrogen exposure and retrograde menstruation [229,230]. Preventive strategies such as physical activity, early contraceptive use, and inflammation control during adolescence may help attenuate hormonal and immune disruptions.

Given the rapid evolution of biomarker research, this section highlights recent high-quality studies—particularly those involving large cohorts or rigorous validation. While not exhaustive, it highlights the genomic, epigenetic, proteomic, and miRNA-based biomarkers under active investigation. Multi-marker combinations are being explored to improve their specificity and clinical relevance for early diagnosis and patient stratification [231,232].

Adenomyosis and endometriosis may coexist, especially in young nulliparous women with dysmenorrhea or endometriomas. External adenomyosis, often underdiagnosed in adolescents, may signal early endometriosis, supporting its inclusion in risk-based screening [233].

Genome-wide association studies (GWAS) have identified susceptibility variants in genes such as *WNT4*, *growth regulation by estrogen in breast cancer 1 (GREB1)*, and *VEZT*, which regulate genital tract development, hormonal signaling, and tissue remodeling [234,235,236,237]. More recently, multi-ancestry GWAS, supported by transcriptome-wide (TWAS) and proteome-wide (PWAS) analyses, have revealed over 45 susceptibility loci, including ancestry-specific signals such as *POLR2M* in African-ancestry populations. These integrative omics approaches have confirmed known genes (e.g., *WNT4*, *GREB1*, *CDC42*) and identified novel ones such as *R-spondin 3 (RSPO3)*, implicating pathways like Wnt signaling, immune modulation, and altered cell migration [238].

In parallel, epigenetic markers—particularly DNA methylation—are under investigation as predictive indicators. *HOXA10* promoter hypermethylation, a key regulator of endometrial receptivity, is found in both eutopic and ectopic tissues, although its sensitivity in early disease remains variable [43]. More recently, cell-free DNA (cfDNA) has emerged as a minimally invasive biomarker. Elevated cfDNA levels—up to 3.9-fold higher in patients—combined with altered methylation in genes such as *ribosomal RNA-processing protein 1 (RRP1)*, *disco-interacting protein 2 homolog C (DIPC2)*, *ubiquitin-specific peptidase 1 (USP1)*, *and DNMT1*, highlight its potential for early-stage detection [239].

Circulating miRNAs are gaining attention as early stratification tools. A five-miRNA panel (miR-17-5p, miR-20a-5p, miR-199a-3p, miR-143-3p, and let-7b-5p) distinguished cases with 96% sensitivity and 79% specificity [240], while a separate six-miRNA signature achieved an AUC of 0.94 for early-stage disease [241]. Additional miRNAs related to inflammation, fibrosis, and hormonal signaling further reinforce their diagnostic potential [242,243].

Proteomic studies offer complementary insights. A large-scale plasma analysis involving over 800 women identified a 10-protein panel (e.g., selenoprotein P, neuropilin-1, complement C9, protein S) that discriminated cases with an AUC up to 0.997 [244]. Another study analyzing pre-diagnostic blood samples found elevated levels of innate immune proteins (e.g., S100A9, ICAM2, TOP1, CD5L), suggesting that immune activation may precede clinical symptoms [245].

Although classical inflammatory markers like IL-8 and CA-125 lack specificity [246], their diagnostic value increases when combined with molecular panels [247,248]. Notably, circulating endometrial cells (CECs) have shown superior diagnostic performance [249,250]. In one study, CECs were detected in 89.5% of patients, outperforming CA-125, especially in early disease [251]. Moreover, heat-shock protein 70 (Hsp70)-positive CECs, as reported by Guder et al., may serve as non-invasive biomarkers, highlighting a possible mesenchymal stem-like origin of lesions [252].

Beyond molecular screening, modifying environmental exposure is essential. Endocrine-disrupting chemicals (EDCs)—including dioxins, phthalates, and BPA—can epigenetically reprogram endometrial tissue, compromising receptivity and implantation [31,253,254,255,256]. Reducing exposure during gestation and adolescence may represent a feasible public health measure risk [257,258,259]. Nutritional interventions, such as diets rich in omega-3 fatty acids and antioxidants may help modulate oxidative and inflammatory pathways implicated in disease initiation [215,260,261].

These biomarker-driven approaches contribute to identifying high-risk individuals before symptom onset and may inform early-stage interventions focused on fertility preservation.

### 5.2. Secondary Prevention: Halting Disease Progression and Preserving Ovarian Function

Secondary prevention in women already diagnosed with endometriosis focuses on halting disease progression and preserving their ovarian reserve before irreversible damage occurs. This approach emphasizes timely interventions guided by molecular biomarkers and personalized strategies [262,263]. A decline in serum AMH is one of the earliest indicators of a reduced ovarian reserve and may precede changes in follicle count or menstrual regularity. Low levels of AMH and elevated FSH have been consistently linked to diminished reserves in affected women [264,265,266]. Insulin-like peptide 3 (INSL3) has emerged as a complementary marker of ovarian stromal aging and functionality, although clinical validation of this is ongoing [267].

Epigenetic changes, particularly hypermethylation of HOXA10, have been associated with progesterone resistance and aberrant Wnt/β-catenin signaling. This methylation pattern is frequently observed in eutopic but not ectopic tissue, correlating with impaired implantation potential. Therapeutic agents such as Gonadotropin-Releasing Hormone (GnRH) analogs, letrozole, and metformin have shown the ability to restore HOXA10 expression, supporting its dual diagnostic and therapeutic relevance [43].

miRNAs are increasingly being used to monitor disease activity and reproductive function. Circulating and peritoneal miRNA profiles reflect inflammatory and angiogenic signaling [268,269,270]. miR-451a, in particular, may support non-hormonal therapeutic approaches by targeting multiple pathways [192]. Serum extracellular vesicles containing lesion-derived miRNAs may inform optimal ART timing [271]. A saliva-based miRNA panel has also demonstrated high diagnostic performance for infertility, enabling non-invasive and personalized fertility assessment [272,273].

Oxidative stress within the ovarian microenvironment is another factor affecting oocyte quality and IVF outcomes. Elevated 8-hydroxy-2′-deoxyguanosine (8-OhdG) and imbalanced glutathione (GSH)/glutathione disulfide (GSSG) ratios in follicular fluid are associated with DNA damage and reduced embryo viability [142,264,274]. Follicular fluid miRNA profiles, which correlate with these oxidative markers, may serve as integrated indicators of oocyte competence [275].

The endometrial microbiome influences implantation, immune tolerance, and endometrial receptivity [276]. *Lactobacillus*-dominant (LDM) profiles are associated with higher pregnancy and live-birth rates, while non-*Lactobacillus*-dominant microbiomes—rich in *Gardnerella*, *Prevotella*, or *Acinetobacter*—have been linked to implantation failure and poor endometrial receptivity [277,278]. In women with repeated implantation failure (RIF), dysbiosis correlates with reduced expression of LIF, HOXA11, and VEGF.

Microbiota-targeted interventions have shown potential in this context. *Transvaginal Lactobacillus* supplementation improved pregnancy rates in ART-failure patients [279], while *Lactobacillus gasseri* reduced lesion growth in animal models by enhancing IL-2 and NK cell activity [280]. Broader studies also implicate urogenital dysbiosis in chronic inflammation and lesion recurrence [281,282]. Although clinical management remains limited, strategies such as probiotics, prebiotics, or microbial transplantation are under evaluation and may contribute to fertility preservation in selected cases. Future research should also focus on characterizing the composition of the upper female reproductive-tract microbiota and elucidating the mechanisms underlying its relationship with endometriosis pathophysiology.

Multi-omic approaches, discussed in Section 5.1, are now applied to dynamically monitor biological activity and therapeutic windows in active disease [283]. These insights contribute to the early identification of therapeutic opportunities in patients at risk of ovarian decline.

### 5.3. Tertiary Prevention: Preventing Recurrence and Individualizing Long-Term Fertility Planning

Tertiary prevention focuses on minimizing the recurrence risk and supporting long-term fertility planning following surgical or medical management. Given the chronic and relapsing nature of the disease, maintenance therapy is essential to sustain remission, protect reproductive potential, and maintain quality of life [262].

Hormonal maintenance therapy, using combined oral contraceptives, progestins, or levonorgestrel-releasing intrauterine systems (LNG-IUS), remains the cornerstone of post-surgical management and has proven efficacy in reducing recurrence and preserving ovarian function [284,285,286,287,288]. These agents suppress ovulation and cyclic estrogen fluctuations, thereby reducing inflammation and lesion reactivation risk.

For patients with ovarian endometriomas, long-term hormonal suppression also mitigates the risks of inflammation-induced follicular damage and surgical insult to the ovarian cortex. Current strategies advocate individualized regimens based on the clinical phenotype, reproductive goals, and treatment tolerance.

In addition to hormonal maintenance, complementary therapies are increasingly being explored to enhance recovery and long-term disease control. Acupuncture has shown benefit in reducing pain following surgery, with a recent meta-analysis reporting significant efficacy over sham treatment [289]. Nutritional and supplemental strategies—including vitamin E, omega-3 polyunsaturated fatty acids (PUFAs), curcumin, and probiotics—exert anti-inflammatory and antioxidative effects. In a randomized controlled trial, vitamin E significantly reduced endometriosis-related pelvic pain (standardized mean difference = −1.63; *p* < 0.00001) [290]. While clinical data are still limited, preclinical studies support the role of ω-3 PUFAs, curcumin, and probiotics in suppressing lesion development and modulating immune responses [291,292]. These integrative strategies may offer supportive benefits in post-operative care and warrant further clinical evaluation.

Incorporating emerging molecular markers of disease activity may further enhance the early detection of recurrence and timely intervention. Molecular diagnostics—such as miRNA panels, angiogenic markers (e.g., VEGF-A, miR-486-5p), and inflammatory mediators (IL-6, IL-8)—may improve recurrence detection and risk stratification [30,168,269].

Fertility preservation remains a core component of tertiary prevention, particularly in women delaying conception. Cryopreservation of oocytes or embryos is increasingly being recommended before repeat surgery, especially in cases of declining AMH, bilateral endometriomas, or diminished ovarian reserves [293,294,295,296]. Longitudinal assessment of AMH, INSL3, and follicular fluid biomarkers (e.g., oxidative stress markers, follicular miRNAs) can help determine the optimal timing for ART or preservation procedures [262,267,297].

Beyond pharmacologic and surgical strategies, patient education and clinician awareness remain essential to improving reproductive outcomes. Early diagnosis, prompt initiation of maintenance therapy, and personalized fertility planning can significantly reduce long-term sequelae. Multidisciplinary models of care involving reproductive endocrinologists, fertility specialists, and pain-management teams are increasingly being advocated for comprehensive patient-centered support.

Table 4 summarizes the key molecular and clinical tools that have been explored for prevention and fertility preservation in endometriosis. Organized by prevention level, it compiles hormonal therapies, molecular biomarkers, complementary strategies, and emerging diagnostics across diverse sample types. The overview highlights the growing potential for stratified and integrative approaches tailored to the biological stage of the disease and individual fertility goals.

Together, these evolving strategies reflect a transition toward individualized, biomarker-guided reproductive care that extends beyond episodic treatment to encompass dynamic, long-term patient monitoring and fertility preservation.

## 6. Conclusions and Perspectives

Endometriosis epitomizes a multifactorial and systemic disorder that transcends pelvic confinement and is rooted in the complex interplay of hormonal dysregulation, chronic inflammation, epigenetic reprogramming, oxidative stress, immune dysfunction, and metabolic remodeling. Far beyond its physical manifestations, the disease disrupts the molecular architecture of both eutopic and ectopic tissues, compromising the ovarian reserve, endometrial receptivity, embryo viability, and maternal–fetal immunotolerance. The present review underscores how estrogen dominance, progesterone resistance, and aberrant activation of signaling pathways (e.g., PI3K/AKT, MAPK) jointly sustain lesion survival, angiogenesis, and fibrotic remodeling while also perturbing fertility-related processes at multiple levels. Importantly, infertility in endometriosis is no longer viewed as a mere consequence of anatomical distortion but is increasingly being recognized as the result of deeply embedded molecular dysfunctions affecting gametogenesis, implantation, and systemic homeostasis.

Advances in high-throughput technologies have unveiled a wide spectrum of therapeutic targets—including ERβ modulation, inflammatory kinase inhibition, epigenetic therapies, immune-checkpoint regulators, miRNA-based strategies, and redox modulators—that collectively pave the way for personalized, fertility-preserving interventions. Several of these approaches, although still in the preclinical or early clinical phase, demonstrate tangible potential to move beyond hormone suppression, offering non-hormonal avenues for patients desiring conception. Simultaneously, the emergence of circulating biomarkers, polygenic risk scores, cfDNA methylation patterns, and proteomic signatures has begun to redefine the landscape of early detection and disease monitoring. This molecular arsenal supports the integration of stratified prevention into clinical care—from risk reduction in asymptomatic individuals (primary prevention) through to fertility-preserving interventions in early disease (secondary prevention) and to recurrence surveillance and reproductive planning after treatment (tertiary prevention). Taken together, these insights advocate for a paradigm shift toward translational precision medicine in endometriosis, where diagnostic, therapeutic, and preventive strategies are molecularly guided and patient-centered (Figure 4). To achieve this shift, clinical implementation will require the development of supportive infrastructure, including the integration of molecular diagnostics and longitudinal biomarker monitoring into assisted reproduction and gynecology care settings. Bridging fundamental mechanisms with clinical practice will not only improve reproductive outcomes but also transform endometriosis care into a proactive, multidisciplinary, and personalized continuum.

## Figures and Tables

**Figure 1 ijms-26-07706-f001:**
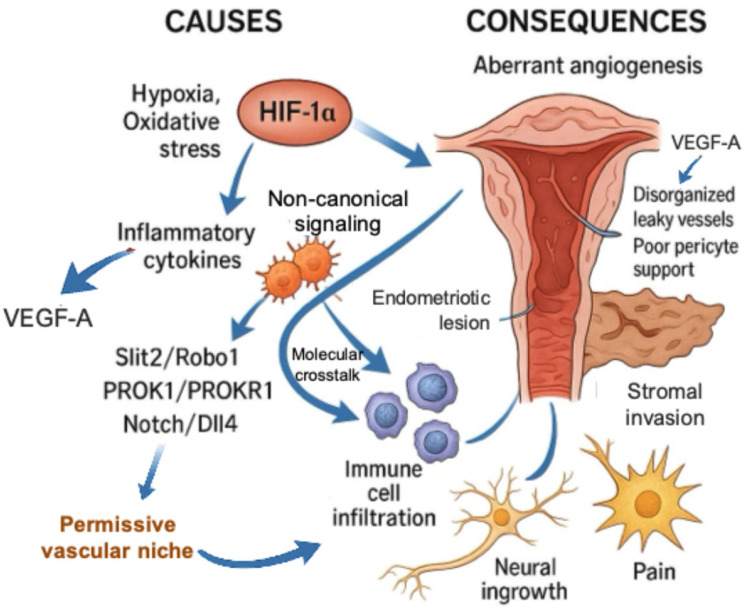
Aberrant vascular remodeling in endometriosis: integration of canonical and non-canonical angiogenic signals. Hypoxia and oxidative stress stabilize HIF-1α, inducing VEGF-A and disorganized neovascularization. Additional non-canonical pathways—including Slit2/ROBO1, PROK1/PROKR1, and Notch/Dll4 suppression—disrupt vessel architecture and branching. The resulting leaky and unstable vasculature facilitates stromal invasion, immune cell infiltration, and neural ingrowth, contributing to a self-perpetuating inflammatory microenvironment.

**Figure 2 ijms-26-07706-f002:**
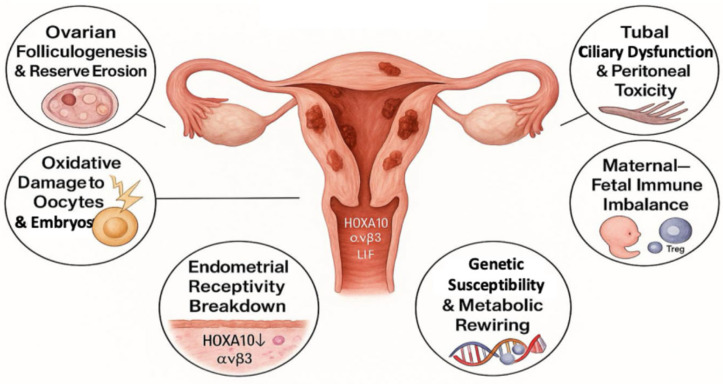
Interconnected mechanisms driving infertility in endometriosis.

**Figure 3 ijms-26-07706-f003:**
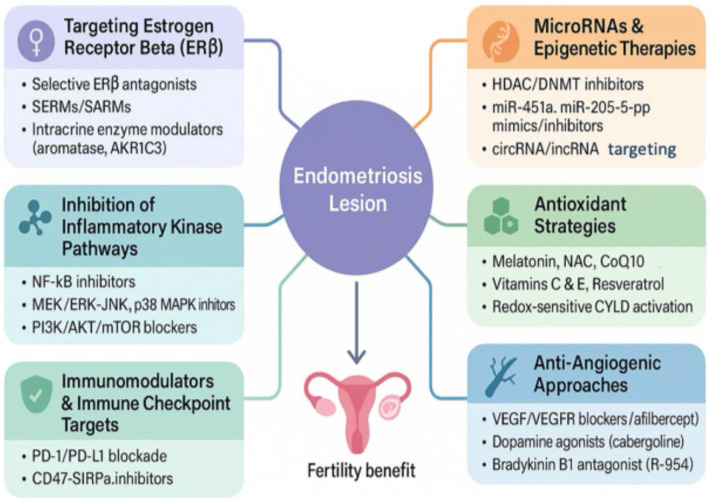
Molecular landscape of non-hormonal therapies in endometriosis.

**Figure 4 ijms-26-07706-f004:**
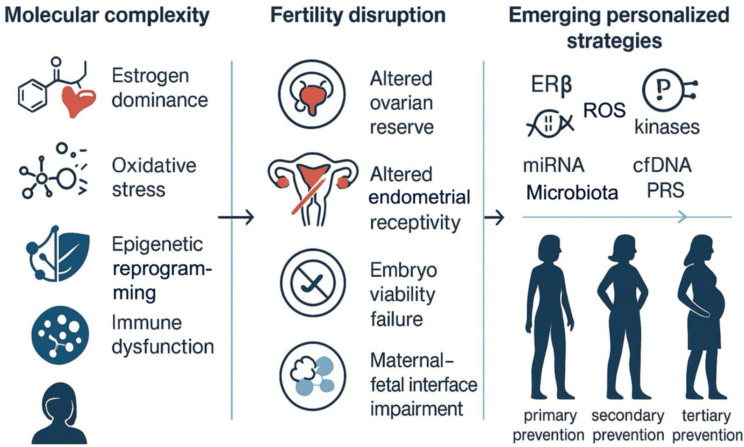
Endometriosis and fertility: molecular complexity and emerging precision approaches. Graphical representation of the multifactorial nature of endometriosis, highlighting key molecular alterations (hormonal imbalance, oxidative stress, inflammation, epigenetic reprogramming), their impact on fertility, and emerging strategies for personalized, fertility-preserving care. Abbreviations: ERβ: estrogen receptor beta; cfDNA: cell-free DNA; PRS: polygenic risk score; ROS: reactive oxygen species.

**Table 1 ijms-26-07706-t001:** Epigenetic and non-coding RNA alterations involved in the pathogenesis of endometriosis and its reproductive consequences.

Molecular Layer	Key Elements	Target Gene/Effects	Molecular Consequence	Biological Impact	References
DNA methylation	CpG hypermethylation	Promoter regions of *HOXA10*, *PR-B*	Silencing of genes crucial for endometrial receptivity	↓ Receptivity, ↑ progesterone resistance, ↓ decidualization	[31,41,42,43,44]
Histone modification	HDACs (class I/II), Sirtuins	Histone deacetylation, chromatin remodeling proteins	Chromatin compaction, ↓ gene expression	Progesterone resistance, ↓ decidualization	[41]
miRNAs	miR-135a/b, miR-29c, miR-194-3p	*HOXA10*, immune-related genes, ECM genes	Post-transcriptional silencing, inflammation signaling	ECM remodeling, ↓ HOXA10, ↓ Receptivity	[45,46,47,48,49,50,51]
lncRNAs	H19	miRNA sequestration	Inhibition of miRNA function	↑ EMT, immune escape	[52,53,54,55]
circRNAs	circ_0007331	miR-200c-3p, *HIF-1α*	Release of HIF-1α expression	↑ Angiogenesis, ↑ steroid resistance	[52,56,57]

Arrows (↑/↓) indicate upregulation or downregulation of the described functional outcomes of the microRNAs (miRNAs), long non-coding RNAs (lncRNAs), and circular RNAs (circRNAs). Abbreviations (in alphabetical order): CpG (cytosine–phosphate–guanine dinucleotide); ECM (extracellular matrix); EMT (epithelial-to-mesenchymal transition); HDAC (histone deacetylase); HIF-1α (hypoxia-inducible factor 1-alpha); HOXA10 (homeobox A10); PR-B (progesterone receptor isoform B).

**Table 2 ijms-26-07706-t002:** Molecular drivers of ECM remodeling, EMT, and fibrogenesis.

Process	Key Molecular Players	Functional Outcome	References
ECM degradation	MMP-2, MMP-9	ECM breakdown, invasion	[100,101,102]
Regulation by TIMPs	TIMP gradients	Directional proteolysis	[103,104,105]
uPA/uPAR system activation	uPA, uPAR, plasmin	MMP activation, fibrin/fibronectin degradation	[106,107,108]
TGF-β–PAI-1–uPAR axis	PAI-1, endothelial senescence	Senescence, early fibrotic changes	[109]
Epithelial-to-mesenchymal transition (EMT)	↓ E-cadherin, ↑ N-cadherin, vimentin	Loss of epithelial identity, invasion	[110,111]
Myofibroblast activation via TGF-β/SMAD	Fibroblasts → myofibroblasts, SMAD pathway	Excessive ECM deposition	[112]
Matrix stiffening by LOX	LOX, collagen I, fibronectin, laminin	Tissue stiffening, ↑ pain	[112]
Angiotensin II/AT1R-induced fibrosis	LOX-like proteins, Ang II, AT1R	Potentiated fibrosis and lesion progression	[107]

Arrows (↑/↓) indicate upregulation or downregulation of the described functional outcomes Abbreviations: Ang II (angiotensin II); AT1R (angiotensin II type 1 receptor); E-cadherin (epithelial cadherin); ECM (extracellular matrix); EMT (Epithelial-to-Mesenchymal Transition); LOX (Lysyl Oxidase); MMP-2 (matrix metalloproteinase-2); MMP-9 (matrix metalloproteinase-9); MMPs (matrix metalloproteinases); N-cadherin (neural cadherin); PAI-1 (plasminogen activator inhibitor-1); SMAD (sma- and mad-related proteins); TGF-β (transforming growth factor beta); TIMP (tissue inhibitor of metalloproteinases); uPA (urokinase-type plasminogen Activator); uPAR (urokinase plasminogen activator receptor); Vimentin (type III intermediate filament protein).

**Table 3 ijms-26-07706-t003:** Promising molecular targets and their impact on infertility in endometriosis.

Therapeutic Strategy	Molecular Target/ Mechanism	Effects on Endometriosis/ Fertility	Development Phase	References
Selective ERβ antagonists	ERβ-mediated inflammation and progesterone resistance	Reduce lesion survival; restore PR-B signaling; preserve ovulation	Preclinical	[17,18,169,170]
Kinase inhibitors (MEK, mTOR, p38, JNK)	NF-κB; MAPK/ERK1/2; PI3K/AKT/mTOR pathways	Decrease lesion growth and inflammation; overcome progesterone resistance	Preclinical	[168,173,175,176,177]
Immune-checkpoint blockade	PD-1/PD-L1 and CD47-SIRPα axes	Restore immune clearance; lower lesion burden; enhance tolerance	Preclinical	[24,179,180,181]
Anti-TNFα therapy	TNFα-driven inflammatory pathways	Reduce pain, inflammation, and lesion size	Preclinical and early clinical	[168,182,183]
Anti-IL-8 therapy (AMY109)	IL-8 mediated chemotaxis and fibrogenesis	Suppress fibrosis and lesion expansion without disrupting cycles	Phase I clinical trial	[185]
Epigenetic modulators (HDACi, DNMTi)	HDACs and DNMTs at HOXA10 and PR-B promoters	Restore decidualization and uterine receptivity	Preclinical	[17,187,188,189,190]
Antioxidants (Vit C/E, CoQ10, NAC, melatonin, resveratrol)	ROS; MAPK and NF-κB activation	Improve oocyte/embryo quality; mitigate oxidative injury and inflammation	Preclinical and early clinical	[196,208,215,216]
Anti-angiogenic/vascular-normalization agents	VEGF/VEGFR2 cascade; dopamine D2 agonists (cabergoline, quinagolide); bradykinin B1	Reduce microvessel density; maintain endocrine function and fertility	Preclinical and Phase 2 clinical trial	[209,210,211,212,213,214,217]
miRNA-based therapy (miR-135a/b, miR-451a, miR-214, miR-205-5p)	Post-transcriptional control of HOXA10, VEGF, CTGF, ANGPT2, CTNNB1, etc.	Reprogram gene networks; suppress angiogenesis, invasion, and fibrosis	Preclinical	[45,192,193,194,195]
SERMs/SARMs/SARDs	ER-subtype balance; intracrine steroidogenesis (aromatase, steroid sulfatase, SLC10A6, AKR1C3)	Re-calibrate local hormones; improve endometrial receptivity	Preclinical and early-to-advanced clinical (Phase II–III) depending on compound	[171]

Abbreviations (in alphabetical order): AKR1C3 (aldo-keto reductase family 1 member C3); ANGPT2 (angiopoietin-2); CD47 (cluster of differentiation 47); CoQ10 (coenzyme Q10); CTGF (connective tissue growth factor); CTNNB1 (catenin Beta 1); DNMTi (DNA methyltransferase inhibitor); ERβ (estrogen receptor beta); ERK1/2 (extracellular signal-regulated kinase 1/2); HDACi (histone deacetylase inhibitor); HOXA10 (homeobox A10); IL-8 (interleukin-8); JNK (c-Jun N-terminal kinase); MAPK (mitogen-activated protein kinase); MEK (mitogen-activated protein kinase kinase); miR (microRNA; mTOR (mammalian target of rapamycin); NAC (N-acetyl-L-cysteine); NF-κB (nuclear factor κ-light-chain-enhancer of activated B cells); PD-1 (programmed cell death protein 1); PD-L1 (programmed death-ligand 1); PI3K (phosphoinositide 3-kinase); PR-B (progesterone receptor isoform B); ROS (reactive oxygen species); SARM (selective androgen receptor modulator); SARD (selective androgen receptor degrader); SERM (selective estrogen receptor modulator); SIRPα (signal-regulatory protein alpha); SLC10A6 (solute carrier family 10 member 6); TNF-α (tumor necrosis factor alpha); VEGF (vascular endothelial growth factor); VEGFR2 (vascular endothelial growth factor receptor 2); p38 (p38 Mitogen-activated protein kinase).

**Table 4 ijms-26-07706-t004:** Biomarker-guided strategies across preventive levels in endometriosis care.

Prevention Level	Biomarker Category	Key Biomarkers/ Interventions	Sample Type	Clinical Objectives	Validation Status	References
Primary	Genomic	GWAS SNPs: WNT4, GREB1, VEZT, RSPO3, CDC42	Blood, tissue	Risk stratification, polygenic scores	Multi-cohort GWAS, TWAS, PWAS	[234,235,236,237,238]
Primary	Epigenetic	HOXA10 promoter methylation	Tissue	Susceptibility receptivity marker	Promising but variable sensitivity	[43]
Primary	Circulating DNA	cfDNA: RRP1, DIPC2, USP1, DNMT1	Blood (serum)	Non-invasive early detection	Preliminary studies	[239]
Primary	miRNA panels	miR-17-5p, miR-20a-5p, miR-199a-3p, miR-143-3p, let-7b-5p; others	Blood, serum, saliva	Early diagnosis, disease discrimination	High sensitivity/specificity, needs expansion	[240,241,242,243]
Primary	Proteomics	Selenoprotein P, Neuropilin-1, C9, Protein S, etc.	Plasma	Early detection, differential diagnosis	Large-scale cohort Validated, AUC up to 0.997	[244,245]
Primary	Cell-based	CECs, mHsp70-positive CECs	Blood	Non-invasive diagnosis	Preliminary evidence	[249,250,251,252]
Primary	Environmental	EDCs: dioxins, BPA, phthalates	Environmental exposure data	Risk reduction, public health intervention	Associative, observational	[31,253,254,255,256,257,258,259]
Primary	Complementary	Vitamin E, antioxidative polyphenols (resveratrol)	Oral, nutritional intake	Anti-inflammatory support, tissue preservation	Preliminary evidence Supported by meta-analyses (vitamin E)	[196,197,204,206,207,215,260,261]
Secondary	Hormonal	AMH, FSH, INSL3	Serum	Ovarian reserve monitoring, fertility potential assessment	Clinically used (AMH, FSH), emerging (INSL3)	[263,264,265,266]
Secondary	Epigenetic	HOXA10 (methylation, restored by GnRH/Letrozole/Metformin)	Tissue	Theranostic marker, progesterone resistance	Experimental, partially reversed pharmacologically	[43]
Secondary	miRNA	miRNA panels	Serum, saliva Follicular fluid	Activity/severity monitoring, ART optimization Embryo competence predictor	Emerging clinical use	[45,192,193,194,195,268,269,270,271,272,273,275]
Secondary	Oxidative stress	8-OHdG, GSH/GSSG	Follicular fluid	Oocyte/embryo viability prediction	Correlational	[142,264,274]
Secondary	Microbiota	*Lactobacillus* profiles, LIF, HOXA11, VEGF	Vaginal swab, uterus	Receptivity marker, microbial intervention	Pilot trials, animal models	[276,277,278,279,280,281,282]
Secondary	Multi-omic	Integrated transcriptomic, proteomic, epigenetic profiles	Various—integrative	Precision-based monitoring	Ongoing integration	[283]
Tertiary	Hormonal maintenance	COCs, progestins, LNG-IUS	Ovarian, endometrial, systemic	Reduce recurrence, preserve ovarian function, suppress estrogen fluctuation	Established in clinical practice	[284,285,286,287,288]
Tertiary	Complementary therapies	Acupuncture, vitamin E, omega-3 PUFAs, curcumin, probiotics	Endometrial, peritoneal	Relieve pain, modulate inflammation, support recovery post-surgery	Preliminary RCTs and preclinical support, ongoing clinical validation	[289,290,291,292]
Tertiary	Molecular	miRNAs, angiogenic markers, inflammatory mediators AMH, INSL3, follicular markers	Serum, tissues, follicular fluid	Monitor residual disease, detect recurrence, guide follow-up	Emerging, under active investigation	[30,168,262,267,269,293]
Tertiary	Functional	Cryopreserved oocytes, embryo viability, clinician–patient planning	Clinical records, cryobank	Reproductive planning, education	Widely adopted in selected cases	[294,295,296]

Abbreviations: 8-OHdG (8-hydroxy-2′-deoxyguanosine), AMH (anti-müllerian hormone), ART (assisted reproductive technology), BPA (bisphenol A), CDC42 (cell division control protein 42), CECs (circulating endometrial cells), cfDNA (cell-free DNA), COCs (combined oral contraceptives), DNMT1 (DNA methyltransferase 1), DIPC2 (DIP2 disco-interacting protein C2), EDCs (endocrine-disrupting chemicals), FSH (follicle-stimulating hormone), GnRH (gonadotropin-releasing hormone), GREB1 (growth regulation by estrogen in breast cancer 1), GSH (glutathione), GSSG (glutathione disulfide), GWAS (genome-wide association study), HOXA10/HOXA11 (homeobox A10/A11), INSL3 (insulin-like peptide 3), LIF (leukemia inhibitory factor), LNG-IUS (levonorgestrel-releasing intrauterine system), mHsp70 (mitochondrial heat-shock protein 70), miRNA (microRNA), Neuropilin-1 (neuropilin 1, a VEGF co-receptor), PUFAs (polyunsaturated fatty acids), PWAS (proteome-wide association study), RRP1 (ribosomal RNA processing protein 1), RSPO3 (R-spondin 3), SNP (single uucleotide polymorphism), TWAS (transcriptome-wide association study), USP1 (ubiquitin-specific protease 1), VEGF-A (vascular endothelial growth factor A), and VEZT (vesicle transporter gene).

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
