# Peer review of "Precision Therapeutic and Preventive Molecular Strategies for Endometriosis-Associated Infertility"

_ijms, 2025, doi:10.3390/ijms26167706_

Round 1
Reviewer 1 Report
Comments and Suggestions for Authors
Inès Limam al. in “Precision Therapeutic and Preventive Molecular Strategies for Endometriosis-Associated Infertility” show a that a molecular arsenal supports the integration of stratified prevention into clinical care from risk reduction in asymptomatic individuals (primary prevention), through fertility-pre-serving interventions in early disease (secondary prevention), to recurrence surveillance and reproductive planning after treatment (tertiary prevention).Collectively, these goals advocate for a paradigm shift toward precision translational medicine in endometriosis.
The review is well written ana interesting.
I consider original the consideration that a insights advocate for a paradigm shift toward translational precision medicine in endome triosis, where diagnostic, therapeutic, and preventive strategies are molecularly guided and patient centered.
The references are appropriate and recent. They support the conceptualizations present in the review.
Authors should add a graphical scheme that summarizes Conclusions and perspectives.
Authors should add a table to better explain what is explained in the paragraph: . From risk reduction to fertility preservation: biomarker-guided strategies in the treatment of domometriosis
Comments on the Quality of English Language
The English could be improved to more clearly express the research.
Author Response
Reviewer 1 comments
Inès Limam al. in “Precision Therapeutic and Preventive Molecular Strategies for Endometriosis-Associated Infertility” show a that a molecular arsenal supports the integration of stratified prevention into clinical care from risk reduction in asymptomatic individuals (primary prevention), through fertility-pre-serving interventions in early disease (secondary prevention), to recurrence surveillance and reproductive planning after treatment (tertiary prevention).Collectively, these goals advocate for a paradigm shift toward precision translational medicine in endometriosis. The review is well written ana interesting.I consider original the consideration that a insights advocate for a paradigm shift toward translational precision medicine in endome triosis, where diagnostic, therapeutic, and preventive strategies are molecularly guided and patient centered.
The references are appropriate and recent. They support the conceptualizations present in the review.
Authors should add a graphical scheme that summarizes Conclusions and perspectives.
Authors should add a table to better explain what is explained in the paragraph: . From risk reduction to fertility preservation: biomarker-guided strategies in the treatment of domometriosis
The English could be improved to more clearly express the research.
Responses
As recommended, we have added a new table (Table 4) at the end of Section 5, summarizing the biomarker-guided strategies discussed across the three levels of prevention, with illustrative examples and clinical implications.
In addition, a graphical summary (Figure 4) has been included to visually synthesize the main conclusions and perspectives.
The manuscript has been carefully revised to improve language clarity and style. Several sentences were rephrased for greater fluidity and precision. All modifications are visible in the revised version of the manuscript.
Reviewer 2 Report
Comments and Suggestions for Authors
Dear Authors,
I read your manuscript with great interest. To improve the article I suggest:
-do a brief overwiew on pathogenic theories (e.g. Sampson’s Theory, Coelomic Metaplasia Theory, Benign Metastasis Theory, Role of Fibrosis, Iatrogenic Theory, Genetic Predisposition Theory
- for the impact on ovarian function you might also find this recent article useful doi: (10.3390/healthcare13080948)
- in Tertiary Prevention section discuss about the complementary therapy to improve the effectiveness of traditional therapy
- add a paragraph on the possible presence of concomitant adenomyosis, even at a young age (it could be useful this recent article doi: 10.3390/diagnostics14212344.)
Best Regards
Author Response
Reviewer 2 comments
Dear Authors,
I read your manuscript with great interest. To improve the article I suggest:
-do a brief overwiew on pathogenic theories (e.g. Sampson’s Theory, Coelomic Metaplasia Theory, Benign Metastasis Theory, Role of Fibrosis, Iatrogenic Theory, Genetic Predisposition Theory
- for the impact on ovarian function you might also find this recent article useful doi: (10.3390/healthcare13080948)
- in Tertiary Prevention section discuss about the complementary therapy to improve the effectiveness of traditional therapy
- add a paragraph on the possible presence of concomitant adenomyosis, even at a young age (it could be useful this recent article doi: 10.3390/diagnostics14212344.)
Best Regards
Responses
- Response: We acknowledge the relevance of these classical theories. However, we chose to focus the review on molecular mechanisms to avoid redundancy with existing reviews and to maintain a concise manuscript. Nonetheless, we added a contextual sentence in the Introduction to acknowledge their historical importance: "Classical theories of pathogenesis—such as retrograde menstruation, metaplasia, and benign metastasis—have laid important foundations, but recent molecular insights have revealed a far more complex, systemic disorder."
- We thank the reviewer for this useful reference, which has been cited in Section 3.1 of the revised manuscript (reference 126).
- We thank the reviewer for this valuable suggestion. A concise paragraph on complementary and integrative approaches have been included in the Tertiary prevention section (page 25, line 1053 of the revised manuscript.
- We have added a discussion on the frequent coexistence of adenomyosis in young women with endometriosis, particularly its implications for tertiary prevention and fertility planning. The suggested reference has been integrated (reference 233).
Reviewer 3 Report
Comments and Suggestions for Authors
This review covers a broad spectrum of molecular mechanisms and emerging therapeutic strategies for endometriosis-associated infertility. The authors provide a highly informative and up-to-date synthesis of hormonal dysregulation, immune dysfunction, epigenetic changes, oxidative stress, and their collective impact on reproductive health. The discussion of precision medicine approaches and preventive strategies is particularly relevant and timely.
However, the manuscript is extremely dense and, while highly informative, may be challenging for readers less familiar with molecular biology. Simplification or restructuring of some sections could enhance readability without compromising scientific rigor. Shorter paragraphs, clearer subheadings, and additional visual summaries may also help to guide the reader through complex content. Some parts of the manuscript should be sumarized in a table. I think the manuscript should be shorter.
Author Response
Reviewer 3 comments
This review covers a broad spectrum of molecular mechanisms and emerging therapeutic strategies for endometriosis-associated infertility. The authors provide a highly informative and up-to-date synthesis of hormonal dysregulation, immune dysfunction, epigenetic changes, oxidative stress, and their collective impact on reproductive health. The discussion of precision medicine approaches and preventive strategies is particularly relevant and timely.
However, the manuscript is extremely dense and, while highly informative, may be challenging for readers less familiar with molecular biology. Simplification or restructuring of some sections could enhance readability without compromising scientific rigor. Shorter paragraphs, clearer subheadings, and additional visual summaries may also help to guide the reader through complex content. Some parts of the manuscript should be sumarized in a table. I think the manuscript should be shorter.
Responses
We sincerely thank the reviewer for this thoughtful and constructive feedback. In response, we undertook a comprehensive revision of the manuscript to improve its overall clarity, organization, and accessibility. The entire text was restructured to ensure smoother transitions, reduce redundancy, and make dense sections more digestible—while preserving the depth and integrity of the scientific content. Paragraphs were shortened for better readability and flow. We also introduced visual elements (Figures 1 and 4) and three integrative tables (Tables 1, 2, and 4) to help guide readers through key concepts and complex data. These efforts were made to enhance comprehension and engagement, especially for readers less familiar with molecular biology, without compromising scientific rigor.
Round 2
Reviewer 3 Report
Comments and Suggestions for Authors
I am satisfied with the current version of the revised manuscript and believe it addresses all the comments and suggestions provided.